

# A spatially downscaled sun-induced fluorescence global product for enhanced monitoring of vegetation productivity

Gregory Duveiller[1], Federico Filipponi[1], Sophia Walther[2], Philipp Köhler[3], Christian Frankenberg[3,4], Luis Guanter[5], and Alessandro Cescatti[1]

[1]European Commission Joint Research Centre, Ispra, Italy
[2]Max Planck Institute for Biogeochemistry, Jena, Germany
[3]California Institute of Technology, Pasadena, CA, USA
[4]Jet Propulsion Laboratory, California Institute of Technology, CA, USA
[5]Universitat Politècnica de València, Valencia, Spain

**Correspondence:** Gregory Duveiller (gregory.duveiller@ec.europa.eu)

**Abstract.** Sun-induced chlorophyll fluorescence (SIF) retrieved from satellite spectrometers can be a highly valuable proxy for photosynthesis. The SIF signal is very small and notoriously difficult to measure, requiring sub-nanometer spectral resolution measurements, which to-date are only available from atmospheric spectrometers sampling at coarse spatial resolution. For example, the widely used SIF dataset derived from the GOME-2 mission is typically provided in 0.5° composites. This paper
presents a new SIF dataset based on GOME-2 satellite observations with an enhanced spatial resolution of 0.05° and an 8-day time step covering the period 2007-2018. It leverages on a proven methodology that relies on using a light use efficiency (LUE) modelling approach to establishing a semi-empirical relationship between SIF and various explanatory variables derived from remote sensing at finer spatial resolution. An optimal set of explanatory variables is selected based on an independent validation with OCO-2 SIF observations, which are only sparsely available but have a high accuracy and spatial resolution.
After a bias-correction, the resulting downscaled SIF data shows high spatio-temporal agreement with the first SIF retrievals from the new TROPOMI mission, opening the path towards establishing a surrogate archive for this promising new dataset. We foresee that this new SIF dataset should be a valuable asset for Earth System Science in general, and for monitoring vegetation productivity in particular. The dataset is available at: https://doi.org/10.2905/21935FFC-B797-4BEE-94DA-8FEC85B3F9E1 (Duveiller et al., 2019).

## 1  Introduction

Mapping and monitoring the spatial and temporal patterns of terrestrial gross primary productivity (GPP) through the use of satellite remote sensing is of paramount interest for vegetation, ecosystem, and climate science. While the rate of terrestrial photosynthesis cannot be directly measured from space, recent research has demonstrated that sun-induced chlorophyll fluorescence (SIF) retrieved from satellite spectrometers can be a highly valuable proxy (Frankenberg et al., 2011). SIF is generally
positively correlated with leaf photochemistry during specific light conditions that are common across the globe and should thus serve as valid proxy for GPP, even though the mechanistic link between the two is complex (Porcar-Castell et al., 2014).





The origin of this signal is the fluorescence of chlorophyll *a*, consisting of a re-emission of absorbed photons at lower energy wavelengths (from 650 to 850 nm, with peaks at approximately 690 and 740 nm). This physical mechanism allows photosynthetic organisms to adjust instantaneously to rapid perturbations in environmental conditions of light, temperature and water availability, before the heat dissipation mechanism of non-photochemical quenching can be triggered (Maxwell and Johnson, 2000).

Chlorophyll *a* fluorescence has been extensively studied in laboratory from the sub-cellular scale up to the leaf for many decades (Baker, 2008), but only recently has it been possible to retrieve it from space-borne remote sensing platforms at global scale. The majority of leaf-scale fluorescence research focused on induction kinetics, e.g. the temporal course of fluorescence (induced by saturating light pulses under various conditions) during the period of light-acclimation of dark-adapted leaves. On the contrary, passive remote sensing from satellite platforms allows only to measure the light-adapted steady-state fluorescence (or SIF). Yet, this has opened the possibility to explore the carbon cycle dynamics in various terrestrial ecosystems, ranging from tropical forests (Lee et al., 2013; Parazoo et al., 2013) to the northern tundra (Walther et al., 2018), passing by agricultural landscapes (Guan et al., 2015; Guanter et al., 2014). It has been used to analyse the response of plants to water availability (Walther et al., 2019) and drought (Parazoo et al., 2015; Yoshida et al., 2015; Sun et al., 2014), but also as an alternative measure to typical remote sensing greenness indices such as NDVI (Normalized Difference Vegetation Index) and EVI (Enhanced Vegetation Index) for vegetation phenology (Joiner et al., 2014; Walther et al., 2016).

Despite the abundance of recent studies, currently there is no operational satellite instrument specifically dedicated to the measurement of SIF. The SIF signal is notoriously difficult to measure from space, as it represents only 1 to 5% of the total reflected radiation in the near-infrared that is detected by a remote sensing instrument (Meroni et al., 2009). Measuring it requires both high spectral resolution and high signal-to-noise ratio, which generally comes at the expense of spatial detail. The first mission specifically designed for this purpose is the European Spatial Agency's Earth Explorer FLEX (FLuorescence EXplorer), due to be launched in 2022. All datasets that are currently available come from missions originally conceived for measuring atmospheric trace gas concentrations. These can be separated among those providing highly precise soundings but with sparse and thus spatially discontinuous samples (e.g. GOSAT, TanSat and OCO-2), and those which do provide a spatially continuous coverage but with less accuracy due to a reduced spectral resolution (e.g. SCIAMACHY, GOME-2 and more recently, TROPOMI). After the initial discovery that such instruments could be serendipitously used to retrieve SIF from space (Frankenberg et al., 2011; Joiner et al., 2011; Guanter et al., 2012), different retrieval methods have been developed to adjust to the various sensors (Joiner et al., 2013; Köhler et al., 2015; Sun et al., 2018; Köhler et al., 2018), each with specific properties such as different spectral fitting windows and cloud filtering procedures, resulting in various distinct SIF datasets.

For a SIF dataset to become truly useful to the Earth System Science community as a proxy for GPP, the following properties should be optimized: (1) the temporal archive should be as long as possible; (2) the revisit frequency should be high (ideally at daily or even sub-daily scale); (3) the geographic extent should be global (ideally gap-free); and (4) the spatial resolution should be fine enough to relate to distinct land cover elements or plant functional types (PFTs). No current individual SIF dataset adequately satisfies these specifications. Retrievals based on GOME-2 (Joiner et al., 2013; Köhler et al., 2015) are perhaps at present those closest to the mark, thanks to their spatially continuous sampling design combined with their temporal





archive running since 2007. However, their spatial resolution is 0.5°(approximately 50 km), which is too coarse for many applications in land science. The situation has changed in 2018 with the operational arrival of TROPOMI, from which SIF can be retrieved from a ground pixel footprint of around 7 km by 3.5 km at nadir (Köhler et al., 2018), but the shallow temporal depth of this data record will preclude its use for many applications for various years to come if no compatible archive is

established.

To remedy the lack of spatial detail, several studies have proposed to enhance the spatial resolution of currently available SIF data. The first of such studies (Duveiller and Cescatti, 2016) leverages on the concept of light use efficiency (LUE) used in GPP modeling (Monteith, 1977; Running et al., 2004) to constrain the spatial re-allocation of GOME-2 SIF values within a $0.5° \times 0.5°$ grid cell. This model assumes that SIF can be estimated as a function of greenness, as described by the NDVI,

which is then down-regulated based on water availability and temperature, characterized respectively by evapotranspiration (ET) and land surface temperature (LST). NDVI, ET and LST are all satellite remote sensing variables that are available at fine scale (e.g. $\leq 0.05°$), can be aggregated to the SIF resolution to establish the relationship over a local spatio-temporal window, and which can then be used to predict SIF at finer resolution. Since then, other studies have adopted entirely data-driven approaches relying on machine-learning techniques to either reconstruct SIF based on fine spatial resolution reflectances from

another satellite (Gentine and Alemohammad, 2018; Li and Xiao, 2019) or to gap-fill spatially sparse OCO-2 data (Yu et al., 2018; Zhang et al., 2018). Compared to these entirely empirical approaches, the Duveiller and Cescatti (2016) method has the particularity that it remains data-driven, as the original SIF signal is preserved at each step, yet the downscaling remains physiologically constrained by the assumptions of a semi-empirical process-based model grounded on theory.

The objective of this work is to present an improved and updated downscaled SIF dataset based on the Duveiller and Cescatti

(2016) methodology. Besides an extension of the archive until the end of 2018, the new dataset has a finer temporal frequency and is constructed from both updated explanatory variables and different SIF retrievals. The optimal model configuration is selected based on a comparison with validation data composed of fine spatial and spectral resolution OCO-2 SIF observations. Finally, the resulting downscaled SIF data is compared to the new TROPOMI retrievals in view of constituting an archive for this promising SIF data stream.

## 2   Material and methods

### 2.1   Explanatory variables at fine spatial resolution

The explanatory variables used to downscale SIF are all retrieved from observations of the same instrument called MODIS (MOderate Resolution Imaging Spectroradiometer) that flies on-board of two sun-synchronous orbiting platforms: Terra (with a descending morning orbit) and Aqua (with an ascending afternoon overpass). The variables used in the original Duveiller

and Cescatti (2016) study, i.e. monthly NDVI, ET and LST from the respective MYD13C1, MOD16 (Mu et al., 2011) and MYD11C3 (Wan, 2008) datasets, are all part of version 5 of MODIS products, which are now deprecated and super-seeded by those of version 6. Here, besides using the new version 6 products, we explore the possibility to generate a downscaled SIF product with an 8-day time step, and thus daily or 8-daily MODIS products are used instead of the monthly products





used in Duveiller and Cescatti (2016). For LST, the 8-daily MYD11C2 product based on the Aqua instrument is used (Wan et al., 2015b) to keep afternoon observations (circa 13:30), but these are complemented by the morning MOD11C2 LST data product from Terra (Wan et al., 2015a) to explore whether earlier morning measurements (circa 10:30 AM) can improve the downscaling performance. Instead of working with the pre-computed monthly or 16-day NDVI products, we decided to use the

BRDF-corrected MODIS reflectance MCD43C4 products (Schaaf and Wang, 2015), which provide daily estimations based on a 16-day moving kernel using the methodology of Schaaf et al. (2002). To ensure a temporal match with the LST product, the day corresponding to the centre of the 8-day LST compositing window is used to select the reflectance values of interest, and then calculate NDVI. Furthermore, other indices that could be better downscaling explanatory variables, such as EVI (Huete et al., 2002), NIRv (Badgley et al., 2017) and NDWI (Gao, 1996), are also calculated based on these reflectances. NDWI is

more specifically expected to be a plausible surrogate for the MODIS ET product, the latter being a highly modelled data product in itself. Finally, to retain a comparable product to the original, the ET MOD16A2 product (Running et al., 2017) is also collected. For all product, the relevant quality flags were used to mask out values of inferior quality. All products were retrieved directly from the NASA LPDAAC (https://lpdaac.usgs.gov/) using the R MODIS package (Mattiuzzi and Detsch, 2018), with the exception of the MOD16A2, which unlike the others is not directly available at the 0.05° spatial resolution, and

was thus aggregated to that grid in using a sliding window of 16 days moving at 8-daily time steps using the Google Earth Engine platform (Gorelick et al., 2017).

## 2.2    SIF data from GOME-2, OCO-2 and TROPOMI

The first source of SIF data is from the GOME-2 instrument on-board of the MetOp-A satellite. The SIF retrievals are acquired around 9:30 local time (at the equator) and at a spectral wavelength around 740 nm. The spatial footprint of these measurements

approximately 40 by 80 km before 2013 and 40 by 40 km after that date, as a consequence of the additional monitoring capacities due to the arrival of MetOp-B. Two retrievals from GOME-2 are used in this paper. The first was developed by Joiner et al. (2013) and is referred to as JJ in this study. Version 25 of this retrieval was used in the original Duveiller and Cescatti (2016) paper, which was super-seeded by the version 27, used here. The second retrieval was proposed by Köhler et al. (2015) and is referred to here as PK. For both JJ and PK datasets, the individual retrievals are filtered to keep only those

with solar zenith angles below 70°, local solar time between 8:00 and 14:00, and effective cloud cover fraction below or equal to 0.5. Then they are gridded into 0.5 cells by taking the mean value over a period of 16 days, in line with the MODIS reflectance products. Since the GOME-2 acquisition time is early in the morning, values need to be multiplied by a daily correction factor to make them comparable with estimates from other sources. Frankenberg et al. (2011) proposed a simple approach to convert the instantaneous SIF to a daily average, which accounts for variations in overpass time, length of day, and solar zenith angle.

For the JJ dataset, such daily correction factor is already provided in the dataset. For the PK dataset it is applied using the implementation used in Köhler et al. (2018) as follows:

$$SIF = SIF(t_m) \cdot \frac{1}{\cos(\theta(t_m))} \cdot \int\limits_{t=t_m-12h}^{t=t_m+12h} \cos(\theta(t)) \cdot H(\cos(\theta(t)))dt \qquad (1)$$





where $\theta(t_m)$ is the solar zenith angle at the time of measurement $t_m$ and the integral is computed numerically in 10-min time steps ($dt$), with the heavyside step function $H$ zeroing out negative values of $\cos(\theta)$. This daily correction factor is applied to the SIF retrievals prior to compositing and gridding.

The second source of SIF data is the OCO-2 platform (Sun et al., 2018). These retrievals are made from soundings over footprints of $1.3 \times 2.25 \text{km}^2$ at nadir, which together create a 10 km wide stripe with a revisit time of 16-days. Retrievals are made at two wavelengths, 757 nm and 771 nm, at 13:00 local time (at the equator). The SIF at both wavelengths are first daily corrected and then, to render them comparable to the GOME-2 SIF at 740 nm, they are combined together using the following formula:

$$SIF_{740\,\text{nm}} = 1.56 * (SIF_{757\,\text{nm}} + 1.8 * SIF_{771\,\text{nm}})/2. \tag{2}$$

The scaling factors were determined based on a reference SIF emission shape derived from leaf-level measurements conducted by Magney et al. (2019). All individual observations between 2015 and 2017 are gridded into a common $0.05°$grid to match the MODIS grid for explanatory variables (and which becomes the grid of the final downscaled SIF product). This results in a validation dataset that contains more than 138 million records distributed across the globe. Each record is further attached to an ancillary information of the major climate zone group in which it falls (tropical, dry, temperate, continental or polar) based on the revisited Koppen-Geiger classification (Kottek et al., 2006) and the dominant vegetation type derived from the European Space Agency's Climate Change Initiative land cover maps (ESA, 2017). Due to their superior spectral and spatial resolution, the SIF retrievals from OCO-2 are here considered as a reference.

The third source of SIF data is from TROPOMI, the single instrument on-board the Sentinel-5 Precursor satellite (Veefkind et al., 2012). These data are available daily since early 2018 with a footprint of 7 km by 3.5 km at nadir. The daily corrected retrieval results with the default filtering as described in Köhler et al. (2018) were aggregated to the $0.05°$grid using the 16-day compositing scheme used for GOME-2. Similar to the other SIF data sets, the filtering consists of thresholds for the fit quality, clouds as well as extremely low/high radiance levels, and high solar zenith angles.

## 2.3 Parametrisation of the downscaling methodology

The downscaling procedure follows 3 steps: (1) an aggregation of the explanatory variables to the coarse spatial resolution; (2) a calibration of the downscaling function over a local spatio-temporal window of coarse spatial resolution data; and (3) the application of the calibrated function to the original explanatory variables at fine spatial resolution to result in a downscaled SIF estimation. The downscaling function always takes the following form:

$$SIF = f(V) \times f(W) \times f(T) \tag{3}$$

where $f(V)$ is a function of a vegetation index $V$, which is down-regulated by two other functions, $f(W)$ and $f(T)$, both outputting a value between 0 and 1 based on indicators $W$ and $T$, which respectively represent water and thermal stresses. A quadratic, a sigmoid and a Gaussian function are respectively used to model $f(V)$, $f(W)$ and $f(T)$, resulting in the following



| Index type | Explanatory variable | parameter | min | init | max |
|:---:|:---:|:---:|:---:|:---:|:---:|
| $V$ | NDVI | $b_1$ | 0.5 | 1 | 1.5 |
| $V$ | NDVI | $b_2$ | 0.1 | 2 | 5 |
| $V$ | EVI | $b_1$ | 0.5 | 1 | 1.5 |
| $V$ | EVI | $b_2$ | 0.1 | 2 | 5 |
| $V$ | NIRv | $b_1$ | 0.5 | 1 | 1.5 |
| $V$ | NIRv | $b_2$ | 0.1 | 2 | 5 |
| $W$ | ET | $b_3$ | 0.05 | 0.1 | 0.5 |
| $W$ | ET | $b_4$ | 1 | 20 | 200 |
| $W$ | NDWI | $b_3$ | 0 | 50 | 500 |
| $W$ | NDWI | $b_4$ | -1 | 0 | 1 |
| $T$ | MOD | $b_5$ | -310 | -295 | -290 |
| $T$ | MOD | $b_6$ | 1 | 10 | 50 |
| $T$ | MYD | $b_5$ | -310 | -295 | -290 |
| $T$ | MYD | $b_6$ | 1 | 10 | 50 |

**Table 1.** Boundary conditions used to initialize the calibration procedure for every local optimization dependent of the explanatory variables used as a proxy for the vegetation $V$ and for the water $W$ and thermal $T$ stresses.

expanded expression:

$$SIF = b_2 V^{b_1} \times \frac{1}{1 + \exp(b_3(b_4 - W))} \times \exp(-0.5\left[\frac{T + b_5}{b_6}\right]^2) \tag{4}$$

The $b_i$ parameters are estimated in the calibration phase for each time step using an adaptable spatial moving window containing the 40 nearest observations around the central pixel. The calibration is done using the Quasi-Newton L-BFGS-B optimization

algorithm (Byrd et al., 1995) implemented in the core R package stats, which allows the setting of a lower and an upper boundary for each parameter.

The difference with the original procedure described in Duveiller and Cescatti (2016) lies in using two different SIF retrievals, $SIF_{JJ}$ and $SIF_{PK}$, and different explanatory variables for $V$, $W$ and $T$. NDVI, EVI and NIRv are three alternative spectral indexes explored for $V$; ET and NDWI for $W$; and morning and afternoon LST, labelled MOD and MYD, are used for

$T$. Various combinations of variables are tested, all using the initial conditions exposed in table 1 for a period coinciding with that of the OCO-2 validation dataset i.e. 2015-2017. The individual values of each downscaled dataset are matched in space and time with those of the OCO-2 dataset and grouped per climate zone and dominant vegetation type.

## 2.4 Quantifying agreement

To quantify the agreement between different sources of SIF data, the $\lambda$ index of agreement proposed by Duveiller et al. (2016)

is used in addition to regular metrics of correlation and bias. This metric quantifies the degree of agreement between two sets of





values, $x$ and $y$, considering both the bias between them and their level of correlation, all within a single number ranging from 0 to 1. It has an added advantage of being symmetric, unlike a typical measurement of goodness-of-fit such as the coefficient of determination, and yet can still be interpreted as a familiar correlation coefficient when there is no bias. The calculation of this index is as follows:

$$\lambda = 1 - \frac{n^{-1}\sum_{i=1}^{n}(x_i - y_i)^2}{\sigma_x^2 + \sigma_y^2 + (\mu_x - \mu_y)^2 + \kappa} \tag{5}$$

where $\mu$ and $\sigma$ represent the mean and standard deviation, respectively. The numerator of the fraction in equation 5 is the mean squared deviation between $x$ and $y$, while the denominator quantifies the maximum deviation this set of points could take. The term $\kappa$ represents the covariance between $x$ and $y$. Including it in the denominator ensures that $\lambda$ does not take negative values when $x$ and $y$ are anti-correlated, but it also unnecessarily inflates $\lambda$ non-linearly. Therefore, $\kappa$ is set to zero when the correlation between $x$ and $y$ is positive, and otherwise takes the value of $\kappa = 2n^{-1}|\sum_{i=1}^{n}(x_i - \mu_x)(y_i - \mu_y)|$.

We also use a variant of $\lambda$ that quantifies only the unsystematic contribution to the agreement, *i.e.* after the systematic bias between $x$ and $y$ is removed. To calculate this $\lambda_u$, the numerator in equation 5 needs to relate only on the unsystematic component of the deviations, instead of the total deviations. The mean squared unsystematic deviation is calculated based on the orthogonal distances $h$ from the principal plane between $x$ and $y$, resulting in the following expression for $\lambda_u$:

$$\lambda_u = 1 - \frac{n^{-1}\sum_{i=1}^{n}h_i^2}{\sigma_x^2 + \sigma_y^2 + (\mu_x - \mu_y)^2 + \kappa} \tag{6}$$

To characterize the principal plane between $x$ and $y$ and obtain the distances $h$, we use an eigen decomposition of the covariance matrix containing vectors $x$ and $y$, as described in Duveiller et al. (2016). This also provides the slope and intercept of the lines between $x$ and $y$ in a symmetric way, *i.e.* the values remain unchanged when $x$ and $y$ are inter-changed.

## 3 Results

### 3.1 Benchmarking the downscaled datasets

The OCO-2 validation dataset is used to identify the optimal combination of explanatory variables and input dataset to produce the enhanced GOME-2 downscaled dataset. Since generating these datasets is computationally expensive, not all combinations were calculated. We instead explore how replacing a single variable at a time affects the results, starting from the initial configuration from the Duveiller and Cescatti (2016) paper: the JJ SIF retrieval downscaled with NDVI, ET and MYD. The results are summarized in Figure 1.

The first variable to substitute is ET with NDWI. NDWI is a much lower-level product than ET, requiring much less assumptions and thus rendering the downscaling independent from other sources of information such as the climate re-analysis data and the eddy-covariance towers used in calibrating ET. Substituting ET for NDWI results in a marked reduction of agreement for the JJ dataset due to a loss in correlation and an increase in bias. For the PK dataset, the drop in correlation is smaller and is accompanied with a reduction of the bias, resulting in an increase of agreement.

The second variable change that is explored is the overpass time of the LST data serving as a proxy for the thermal stress. Replacing the afternoon overpass (MYD) used initially with the morning one (MOD) could be relevant as it is closer to the overpass time of the GOME-2 instrument. On the other hand, data from the morning MODIS instrument may be of lower quality as the instrument is older. This analysis is done only on PK data and shows an increase in both the correlation (marginally) but

also in the bias (more substantially), and as a result this reduces the overall agreement.

The third variable to exchange is the vegetation index from NDVI to EVI and NIRv. For both JJ and PK retrievals there is a progressive increase in correlation and agreement with the best results stemming from the use of NIRv. Replacing NDVI with EVI or NIRv evens reduces slightly the bias for JJ, whilst it marginally increases for PK. For JJ, the loss of agreement due to the replacement of ET by NDWI appears to be considerably mitigated by the use of NIRv instead of NDVI.

In view of these results, a single common configuration of explanatory variables consisting of NIRv, NDWI and MYD is selected for downscaling both JJ and PK retrievals over the longer time period from 2007 to 2018. Both of these downscaled products are made available together as outputs of this study, and will be referred to as a single downscaled SIF dataset containing two separate products (Duveiller et al., 2019).

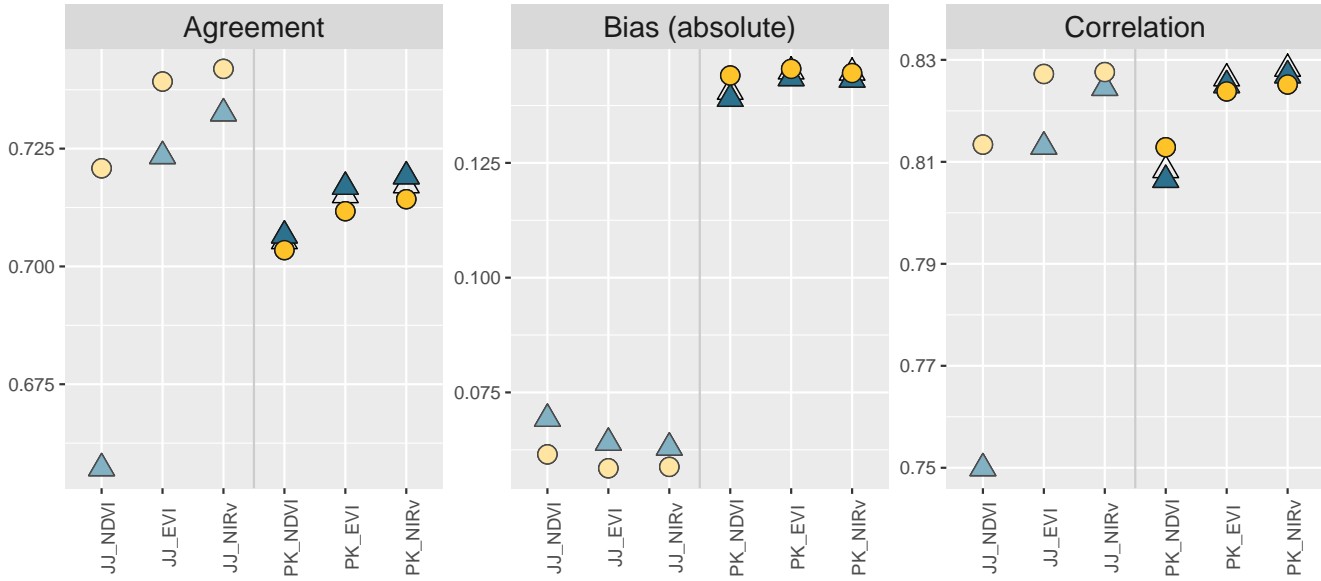

**Figure 1.** Exploring the agreement, absolute bias and correlation between the OCO-2 validation dataset and downscaled GOME-2 SIF based on different retrievals (PK for Köhler et al. (2015) or JJ for Joiner et al. (2013)) and using different explanatory variables (NDVI, EVI or NIRv for $V$). Not all combination were calculated. Yellow circles indicate the use of ET as a proxy for hydric stress ($W$), while triangles represent the use of NDWI instead. Empty triangles correspond to a subset in which the proxy for thermal stress ($T$) is changed from MYD to MOD. Each metric is calculated based on all available samples within the period 2015-2017.

To delve further in the details, Figure 2 compares how both JJ and PK retrievals fare with respect to OCO-2 when downscaled

either with the old configuration of explanatory variables or the new one, but disaggregating the results for different vegetation





types within distinct climate zones. Each graph plots the index of agreement of a given downscaled product with OCO-2 against the index of agreement of the other downscaled product with OCO-2, so that for each point, if it falls on a given side of the 1:1 line, the product of that side has higher agreement with the references. The first panel (Figure 2a) shows that, with the original explanatory variables of NDVI, ET and MYD, the JJ retrieval agrees more with the OCO-2 dataset than the PK retrieval, as the latter is generally affected by a bias. However, some vegetation types in the tropics, such as croplands, grasslands and evergreen broadleaf forests, do have higher agreement with OCO-2 in the PK product, and this is accentuated with the new downscaling methodology (Figure 2c and d). The new methodology only marginally improves the JJ prodcut with respect to the original downscaled SIF data (Figure 2b). The final panel (Figure 2d) provides an inter-comparison of the two downscaled products made available here, potentially guiding users to prefer one or the other depending on the type of climate or the dominant vegetation cover of their area of interest.

To resume the outcome of this benchmarking with respect to the OCO-2 validation dataset, Figure 3 illustrates how both SIF retrievals fare with the new set of downscaling explanatory variables compared both to the old downscaling method and to SIF at the original spatial resolution. For both retrievals, the improvement in agreement between GOME-2 and OCO-2 is due more to the actual spatial downscaling than to the choice of the downscaling variables. The original PK retrieval shows higher agreement than the original JJ retrieval due to higher correlation despite a larger bias, but the downscaling procedure improves both to very similar levels of agreement and correlation.

### 3.2 Exploration of the data

As both PK and JJ downscaled products have broadly similar spatio-temporal patterns, for the sake of brevity and simplicity we are going to focus the rest of this section on only one of the two: PK. The dataset spans from 2007-01-21 until 2018-12-31 with an 8-daily revisiting frequency. To provide an overview of the data, a selection of spatio-temporal subsets are displayed in Figure 4. These represent chrono-sequences of various areas around the globe with a sub-sampled revisit frequency to accentuate the seasonal dynamics of the signal. The rise and fall of productivity in seasonal vegetation is clearly visible in all cases, ranging from the surroundings of the European Alps (A), the East of the Andean Cordillera (B), and even areas around the African Great lakes (C). The impact of agriculture is particularly evident in the irrigated areas of the Indo-Gangetic plains (D) and within the U.S. Corn belt (E). Figure 4 also provides a glimpse of the data gaps that can occur over mountainous areas and lakes, and during rainy periods, which can occur either due to missing coarse scale SIF input data or missing fine scale values of explanatory variables.

To resume the SIF patterns across the globe, Figure 5 provides a map of the maximum SIF value encountered over the entire dataset. This provides a largely gap-free representation of the maximal productivity of terrestrial ecosystems during the 2007-2018 period. The highest values are encountered in intensive agricultural areas of South America, North America and North-Eastern China.



## 3.3 Inter-comparison with TROPOMI

The PK downscaled product overlaps with TROPOMI during the period from 2018-03-18 until 2018-12-23. These two instruments are sounding vegetation in potentially different physiological conditions due to the particular overpass times of their respective satellite platforms (morning for GOME-2 and midday for TROPOMI). While this discrepancy may be mitigated by the daily correction factor and the light-use efficiency downscaling procedure, it may also be compounded by other factors, and a proper inter-comparison is warranted before attempting to establish an archive for TROPOMI based on the downscaled GOME-2 values. For all common records over this period the temporal agreement is thus quantified and mapped in Figure 6a. The $\lambda$ metric takes on high values in the Northern temperate zones, particularly over cultivated areas such as the U.S. corn belt and north-eastern China, where high maximum SIF values are observed in Figure 5. Relatively high agreement ($0.5 \leq \lambda \leq 0.75$) in many highly seasonal areas, such as the Sahel, while areas with no seasonality such as deserts and tropical forests have very low agreement ($\lambda \leq 0.25$). This suggests that a systematic bias is largely responsible for the disagreement, which can be confirmed by mapping the fraction of systematic deviations over total deviations in Figure 6b. This fraction is elevated everywhere, and particularly over South America where the South Atlantic Anomaly of the magnetic field is known to deteriorate the quality of the GOME-2 SIF retrievals (e.g., Köhler et al. (2015)). When the systematic component is ignored using the unsystematic index of agreement $\lambda_u$, the map in Figure 6c reveals that the agreement between both data streams is high. Parts of lower agreement remain in tropical forests and deserts, but even these generally have $\lambda_u \geq 0.75$.

To further illustrate the compatibility of our new downscaled GOME-2 SIF product with TROPOMI SIF retrievals, Figure 7 shows latitudinal profiles of median SIF for different time slices. The systematic bias between both products is evident and relatively consistent. When the bias is corrected using the slope and intercept values obtained at pixel-level based on the common overlapping time series between the downscaled GOME-2 PK SIF and TROPOMI SIF, the resulting median latitudinal profiles closely match each other (reducing the root mean square deviations from 0.188 to 0.0328 mW/m$^2$/sr/nm).

## 4 Discussion

The newly downscaled SIF dataset presented here should be of general interest for Earth System Science, and more particularly for those studying vegetation dynamics over terrestrial ecosystems. We foresee that it could become a valuable asset to better calibrate dynamic global vegetation models (DGVMs), such as those used in TRENDY (Sitch et al., 2015) and which serve as a baseline to establish the yearly global carbon budget (Le Quéré et al., 2018). This SIF dataset could also be used with data-driven approaches, such as those behind widely-used products such as FLUXCOM (Jung et al., 2018) and GLEAM (Miralles et al., 2011), to provide enhanced datasets of the important variables such as GPP and ET.

The comparison of the downscaled PK product with TROPOMI SIF retrievals for the current overlap period shows promising results towards creating a surrogate archive for TROPOMI extending back until early 2007. The discrepancies between both appear to consist in a systematic bias that may originate from a combination of various reasons, ranging from differences in the retrieval approach, sun-surface-sensor geometries, acquisition times and post-filtering. However, correcting this bias empirically results in large agreement between the two data sources. The spatialized pixel-wise coefficients (slope and



intercept) to rescale the GOME-2 downscaled PK SIF retrievals unto the TROPOMI values are also provided along with this dataset. Having been extracted using an eigen decomposition as described in (Duveiller et al., 2016), these coefficients for this linear bias-correction are reference-agnostic, i.e. neither dataset source (GOME-2 or TROPOMI) was explicitly chosen as a reference, and are thus symmetric.

The resulting downscaled products still contains gaps. As mentioned before, gaps can occur due to missing data of either the input SIF or the explanatory variables. Another reason for data gaps occurs over islands or peninsulas, which are areas were there is an insufficient number of neighbouring grid cells to perform the downscaling operation. These gaps could be filled in various ways with different levels of complexity, ranging from statistically smoothing the time series to coupling them with a model such as SCOPE (van der Tol et al., 2009) in a data assimilation system (Lewis et al., 2012). To allow users to have the
maximum level of flexibility, we have here chosen not to do any gap-filling beyond what is already being done using the spatial weighted smoothing included in the original downscaling methodology.

Revisiting the downscaling approach from Duveiller and Cescatti (2016) created an opportunity to explore the use of distinct retrievals and different input variables. It emerges that the factor contributing most to the improvement is the downscaling procedure itself, rather than the choice of the retrieval or that of the explanatory variables. This highlights the benefit and
rationale of seeking to have information coming from a finer spatial support, which is more adequate to characterise the spatial fragmentation of terrestrial ecosystems. The JJ retrieval, which is known to be noisier and with a smaller bias than PK, benefited particularly from the downscaling procedure, probably due to the embedded spatial smoothing step. Regarding the change in explanatory variables, the more important change comes from substituting NDVI by NIRv, which has indeed been shown to be highly correlated to SIF (Badgley et al., 2017). That improvement partly enabled us to tolerate the replacement of ET with the
reflectance-based index NDWI, rendering the output fully independent from eddy-covariance flux-towers and thus of ground-based GPP measurements. The change in the timing of the LST estimation appears marginal based on the analysis using the PK dataset, and it was thus decided to keep the afternoon overpass as it comes from a younger satellite that may be available for longer. However, the effect of LST timing could further be investigated on the JJ retrieval in the future.

The downscaling function based on light use efficiency theory used here could further be fine-tuned to further increase
the performance. In this study and for the present version of our enhanced downscaled SIF dataset, we made a deliberate choice to maintain the overall structure of this function in order to stay compatible with the original Duveiller and Cescatti (2016) work. A possible refinement could come by including information on the incoming photosynthetically active radiation at the surface, if this can be obtained at the fine spatial resolution necessary for downscaling (e.g. Ryu et al., 2018). Another improvement should come from actually considering the fraction of escaping SIF photons relative to the total SIF emitted by
the whole canopy, otherwise known as the escape ratio (Ryu et al., 2019). This would imply using some knowledge of the canopy structure of the vegetation type in question, which may come from estimations of the clumping index based on multi-angular satellite observations (Jiao et al., 2018). Such improvements should be explored and considered in future developments of the present dataset.

The present approach to downscale SIF has some distinctive characteristics with respect to those proposed in other recent
studies (e.g. Gentine and Alemohammad, 2018; Yu et al., 2018; Zhang et al., 2018). First, the downscaling is done within some





physiological constrains imposed by light use efficiency theory, rather than using a purely empirical machine learning approach. This should ground the downscaled values within limits of plausibility and further allow for a certain degree of extrapolation. Yet, the present approach remains data-driven, as the model only disaggregates the SIF signal in space, but does not alter its mean value at a given time and location. The downscaling is also done within a regionalized context, using local moving windows in space (40 nearest pixels) and time (16-days), which ensures a calibration that is tailored to local conditions. Finally, the approach also uses a particular weighted average smoothing using an ensemble of $3 \times 3$ sets of calibrated parameters (for details see Duveiller and Cescatti, 2016) that removes tiling artifacts and partially fills gaps where original SIF retrievals are deemed to be too noisy. Despite these differences, a full inter-comparison between downscaled or reconstructed SIF datasets and a benchmarking with a common and independent reference (such as GPP from flux-towers) should be considered to guide future algorithmic developments and consolidate our capacity to estimate SIF and GPP from space.

## 5 Conclusions

This paper presents a new daily-corrected SIF dataset with a spatial resolution of 0.05° at 8-day time steps for the period 2007-2018 based on two different retrievals of GOME-2 satellite observations. Validation with OCO-2, an independent instrument capable of estimating SIF at finer resolution with a very sparse sampling scheme, has served to identify an adequate combination of explanatory variables to reach this result. A comparison with SIF from the new TROPOMI mission indicates that this downscaled SIF could serve as an archive after a pixel-wise bias correction. As such, we foresee that this new SIF dataset should be a valuable asset for Earth System Science in general, and for monitoring vegetation productivity in particular.

This work also holds promise beyond the simple adjusting of GOME-2 towards TROPOMI. First, it could serve to produce a prior SIF dataset that could be used to optimize the SIF retrievals from the future FLEX mission. The downscaling framework could further be used to downscale TROPOMI retrievals to the spatial resolution of FLEX (∼300 m) by using explanatory variables from instruments on-board of Sentinel-3. FLEX retrievals could even be downscaled to decametric spatial resolution by leveraging on combinations of multi-spectral and thermic instruments including Sentinel-2, Landsat-8 and the potential ESA candidate mission called High Spatio-Temporal Resolution Land Surface Temperature Monitoring (LSTM). Finally, the framework and current dataset could also be adapted towards exploring geostationary satellite data that will be able to provide optimized sub-daily information of plant status.

## 6 Data availability

The dataset of daily corrected downscaled SIF is available in the following repository:

Duveiller et al. (2019): Downscaled-GOME2-SIF. European Commission, Joint Research Centre (JRC) [Dataset] doi:10.2905/21935FFC-B797-4BEE-94DA-8FEC85B3F9E1 PID: http://data.europa.eu/89h/21935ffc-b797-4bee-94da-8fec85b3f9e1.



The files are grouped by year in distinct NetCDF files for each of the two GOME-2 retrievals (JJ and PK). The product it distributed in an equirectangular projection with a pixel size of $0.05°$. The temporal coverage spans from 2007 until 2018. The temporal sampling of the product is 8 days. However, every record is based on SIF input data retrieved over a 16-day moving window. This results in a certain amount of temporal auto-correlation, as the 16-day window moves every 8 days, leaving an

overlap of 8 days in each successive record. The day reported in the NetCDF file corresponds to the 9th day of the 16-day retrieval period to match the MODIS convention used in the MCD43C4 product. Along with the dataset, we also provide in a separate file the slope and intercept values at pixel-level to allow users to rescale the downscaled PK GOME-2 values to TROPOMI estimates.

*Author contributions.*    G.D., F.F. and A.C. designed the study. S.W., P.K. and C.F. prepared and provided input data and guidance on how

to use it. F.F. and G.D. processed the data. G.D. analyzed the data and made the figures. G.D. wrote the text with contributions from all the authors.

*Competing interests.*    The authors declare they have no competing interests.

*Disclaimer.*    The data is provided as is with no warranties.

*Acknowledgements.*    We thank Dr. Joanna Joiner for valuable discussions regarding her GOME-2 SIF retrieval and how to use it in the context

of this study.

G. Duveiller is funded at the European Commission Joint Research Centre by the Copernicus-2 administrative agreement (nr. 5054) supported by DG-GROW.

Most of the heavy calculations were done using the JEODPP platform (Soille et al., 2018).



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

**Figure 2.** Inter-comparison of the performance of different downscaled products disaggregated per major vegetation type and climate zones. The downscaled products are based on either the PK or JJ retrievals combined with either the old set of explanatory variables (NDVI, ET and MYD) or the new one (NIRv, NDWI and MYD). Each graph plots the agreement of a given downscaled product with OCO-2 over the period 2015-2017 against the agreement of another product with OCO-2 for the same period. The overall bias (B), correlation (r) and agreement index (L) are reported for at the corresponding corner of each graph, with numbers in bold highlighting the better values in each pairwise comparison.

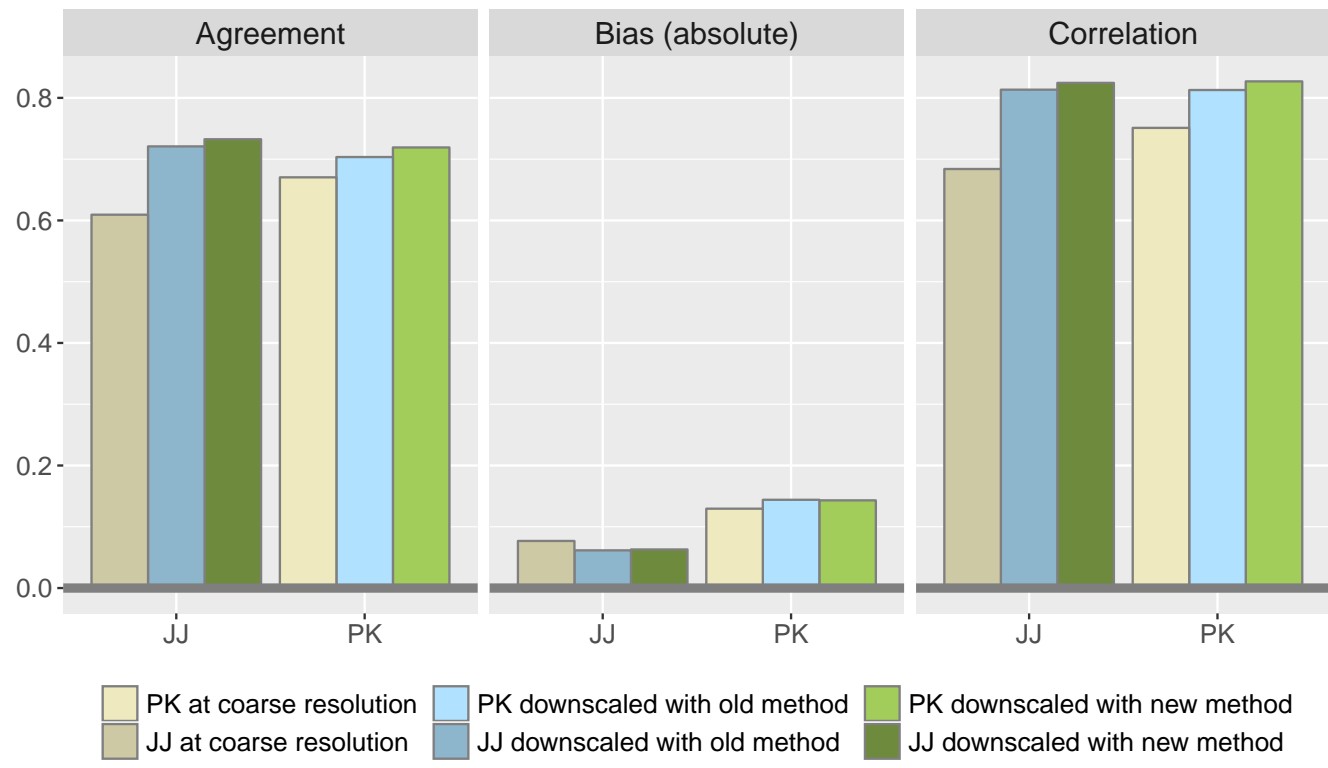

**Figure 3.** General benchmarking of the performance of GOME-2 SIF downscaling versus OCO-2 validation measures for the period 2015-2017. The old downscaling method refers to that based on the original explanatory variables used in Duveiller and Cescatti (2016), i.e. NDVI, ET and afternoon LST, while the new method refers to the best explanatory variables identified in this study, i.e. NIRv, NDWI and afternoon LST. PK refers to the GOME-2 SIF retrieval proposed by Köhler et al. (2015), while JJ refers to that by Joiner et al. (2013). The agreement refers to the index of agreement $\lambda$ proposed by Duveiller et al. (2016).





**Figure 4.** Selection of spatio-temporal subsets of the newly downscaled PK SIF product. Each box covers a region of $8° × 8°$ covering parts of: (A) the European Alps and their surroundings; (B) Bolivia; (C) the African great lakes, (D) the Indus valley and the Gangetic plains, and (E) the US corn belt (the locations of these areas are shown on Figure 5). The corresponding time is mentioned above each image. Although the temporal frequency of the dataset is 8-daily, only an image every 24 days is presented here to accentuate the seasonal dynamics. Notice also that the 5 time series are not synchronized. Grey areas indicates lack of data.

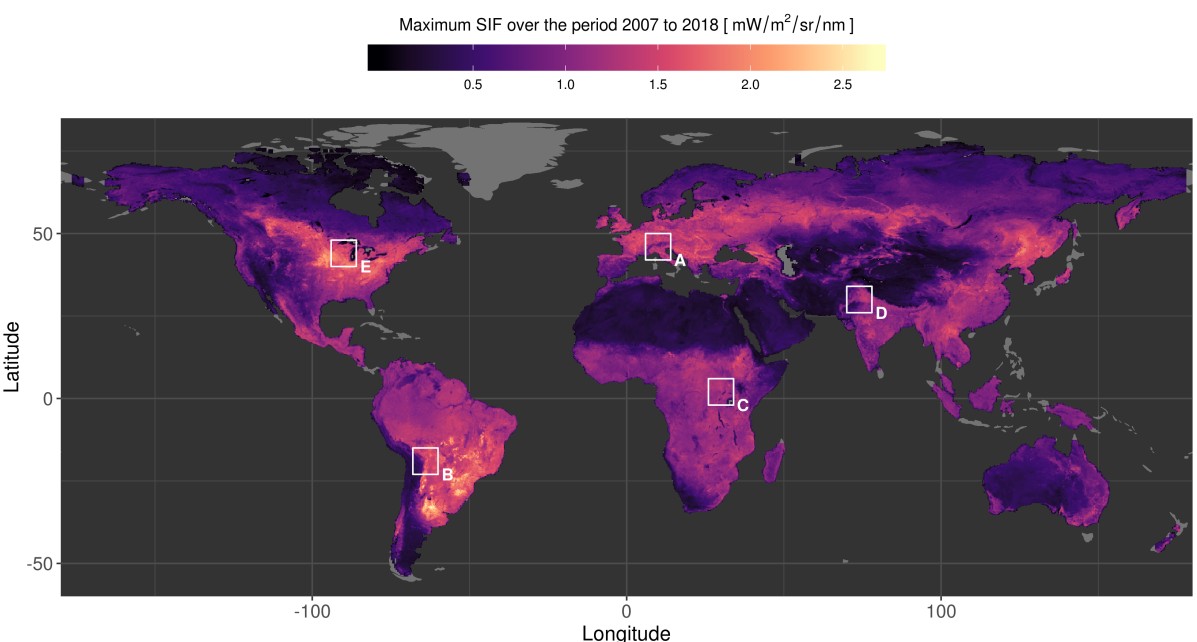

**Figure 5.** Maximum value of SIF over the entire new PK downscaled product. Values are in mW $m^{-2}sr^{-1}nm^{-1}$, the spatial resolution is 0.05°and the temporal ranges spans from early 2007 until end of 2018. The white boxes represent the zones illustrated in Figure 4).



**Figure 6.** Agreement between the downscaled PK SIF product and TROPOMI SIF retrievals for the period from 2018-04-16 until 2018-12-23. The top panel shows the total agreement using the $\lambda$ metric, based on all deviations irrespective on whether these are systematic or non-systematic deviations. The middle panel shows the ratio between systematic and total deviations. The third panel shows the agreement based only on the unsystematic component, $\lambda_u$, in which any disagreement due to a systematic bias is removed.

**Figure 7.** Comparison of latitudinal profiles of TROPOMI SIF, downscaled GOME-2 PK SIF, and the same downscaled GOME-2 SIF but bias-corrected based on the slope and intercept obtained at pixel-level over the overlapping time series.