# Peer review of "A spatially downscaled sun-induced fluorescence global product for enhanced monitoring of vegetation productivity"

_Earth System Science Data, 2019_

## Referee Comment (RC1) · Anonymous Referee #1 · 1 Oct 2019

This paper proposed a method to spatially downscaled the coarse sun-induced fluorescence (SIF) datasets to a finer resolution. The method proposed in this study is sound and I also agree the dataset should be useful for the community of the Earth System Science.

In fact, VIs including NDVI, NIRv and EVI used in this study definitely have a good correlation with SIF, especially at a weekly or longer time resolution. Thus, the high correlation coefficients in the text are under expectation. But, I don't think the VI-derived SIF data product has an advantage in predicting "invisible" phenology of photosynthesis. The performance of this downscaled "SIF" dataset still depends on how well it can

detect changes in vegetation greenness. However, the carbon uptake by green leaves may change throughout the season. Thus, the downscaled dataset may not provide information beyond greenness. The authors should discuss it in the text.

---

## Referee Comment (RC2) · Anonymous Referee #2 · 3 Dec 2019

This manuscript by Duveiller et al. presented a new SIF dataset that is developed based on a previously published method (Duveiller et al., 2016 RSE). In this manuscript, the results from combination of multiple input variables were compared, as well as training against two GOME-2 SIF dataset (PK or JJ). The authors used OCO-2 SIF and TROPOMI SIF as reference for the comparison. The manuscript is clearly written, and the updated dataset seems to improve to some extent as compared to the previous one with extended temporal coverage.

However, I do have some comments for the authors to consider and possibly improve this dataset. 1. The SIF light use efficiency model: I have two concerns for this model.

First, SIF has a unit of energy flux, and in LUE models, the energy input is also an important variable. This model developed by the authors does not include an energy flux term, e.g. PAR. This could have limited effects if the authors assume that the cloud cover is homogenous within each local spatio-temporal window, but how much confidence do we have for this prerequisite should be discussed. Second, the authors used a sigmoid function of ET or NDWI to assess the water stress on vegetation, to me, this is problematic. The changes in ET or NDWI is strongly affected by the vegetation status, i.e., vegetation coverage or vegetation index. For example, during the green-up period, both ET, GPP enhanced as a results of vegetation greening. However, lower ET or NDWI values in the earlier period does not indicate that vegetation is more water limited. Normalization is necessary to use these variables to assess water stress. 2. As a journal specifically targeted at publishing dataset, I would suggest the authors provide enough details in the method for generating this SIF dataset. For example, in the last paragraph of section 2.4, how does the eigen decomposition work is not clear. The spatial and temporal window sizes are also not informed. Although the original method is described in details in a previous publication, since this journal is a data journal, the readers should gain enough understanding of the dataset without referring to other papers. Otherwise, this paper is more like an addendum or update to the previous paper. 3. The author mentioned that the dataset has spatial and temporal gaps in some areas due to the missing values for the GOME-2 SIF or the predictor variables. Would there be a method to solve this issue? The author mentioned about the potential usage for this dataset, however, the gaps would limit these potential applications. 4. The JJ SIF dataset shows an abnormal decreasing trend due to the sensor degradation (Zhang et al., 2018), how about the PK dataset? Since the downscaling are based on these two datasets, this needs to be further discussed. How does the algorithm deal with this issue, if the problem still exists, this needs to be informed and the users should be cautious for trend analysis using this dataset.

Below are some detailed comments: P3 L2 "land science"-> "Earth science"? P6 L2, it would be good to explain the meaning of these b parameters a little bit, it will better help

readers understand the ranges used in Table 1. P8 L4, do you have any references to support this? The two instruments should be exactly the same. P8 L5, why only on PK data? What about JJ data? P10 L30: I don't think so, this is just a high-resolution SIF dataset, it cannot be compared directly with TROPOMI SIF, for example, you cannot use downscaled SIF for year 2017 and compared with TROPOMI SIF for 2018 to detect changes.

---

## Author Response (AR1)

Please find below the remarks from the reviewes (in black), followed by our responses in (blue).

Reviewer 1

This paper proposed a method to spatially downscaled the coarse sun-induced fluorescence (SIF) datasets to a finer resolution. The method proposed in this study is sound and I also agree the dataset should be useful for the community of the Earth System Science. In fact, VIs including NDVI, NIRv and EVI used in this study definitely have a good correlation with SIF, especially at a weekly or longer time resolution. Thus, the high correlation coefficients in the text are under expectation. But, I don't think the VI-derived SIF data product has an advantage in predicting "invisible" phenology of photosynthesis. The performance of this downscaled "SIF" dataset still depends on how well it can detect changes in vegetation greenness. However, the carbon uptake by green leaves may change throughout the season. Thus, the downscaled dataset may not provide information beyond greenness. The authors should discuss it in the text.

We are glad the the reviewer agrees with us that the dataset should be useful for the Earth System Science community at large. However, regarding the comment about how the downscaled SIF data may not provide information beyond greenness, we believe the reviewer might have missed a point regarding some choices in the methodology have been taken precisely to avoid this shortcoming. The downscaling is based on a semi-empirical light-use efficiency framework that is *locally* calibrated both in space and time. At each time step, the spatial disaggregation of the information of every SIF pixel is done based on the spatial distribution of the finer spatial scale pixels of NIRv, NDWI and LST over that single time window. This is done independently at every time step. As a result, the information contained in the downscaled time series is still following the same general pattern as that of the original SIF time series, and not that of the greenness which is used to downscale it. Therefore, if we assume that SIF contains more information than what is available from the 'greenness', which is the claim many publications have made about SIF in the past years, then the downscaled SIF we provide the same information. This local adjustment of the downscaling is actually what differentiates our work methodologically from that of others who used fixed models and basically rescale SIF to greenness (e.g. Gentine & Alemohammad, 2018). We have now discussed this in more detail in the revised manuscript.

Reviewer 2

This manuscript by Duveiller et al.presented a new SIF dataset that is developed based on a previously published method (Duveiller et al., 2016 RSE). In this manuscript, the results from combination of multiple input variables were compared, as well as training against two GOME-2 SIF dataset (PK or JJ). The authors used OCO-2 SIF and TROPOMI SIF as reference for the comparison. The manuscript is clearly written, and the updated dataset seems to improve to some extent as compared to the previous one with extended temporal coverage.

However, I do have some comments for the authors to consider and possibly improve this dataset.

1. The SIF light use efficiency model: I have two concerns for this model.

First, SIF has a unit of energy flux, and in LUE models, the energy input is also an important variable. This model developed by the authors does not include an energy flux term, e.g. PAR. This could have limited effects if the authors assume that the cloud cover is homogenous within each local spatio-temporal window, but how much confidence do we have for this prerequisite should be discussed.

We actually had already mentioned this in the original version. A good reason why PAR is not included directly is that there are no direct estimations of surface in-coming PAR derived from MODIS products that are ready to be used for downscaling. We had mentioned a possible simulated product by Ryu et al, (2018), but this needs to be investigated. In a way, our approach already includes indirectly a proxy for PAR (to some extent) by the intermediary of the LST, which should be highly correlated to PAR. But we agree that there is room for improvement and this is noted in the manuscript.

Regarding the homogeneity of the cloud cover within the window, we could argue that in principle all processing is based mostly on cloud-free observations, as that is when satellite instruments can sample the ground. However, we know that the products we use have different capacities in detecting (and filtering) clouds for various reasons: MODIS has a finer resolution and thus can see smaller clouds, the SIF retrieval is less sensitive to cloud cover, the platforms have different orbit passing times, and thus are sensitive to different clouds. As suggested by the reviewer, we have added a discussion of this in the revised manuscript.

Second, the authors used a sigmoid function of ET or NDWI to assess the water stress on vegetation, to me, this is problematic. The changes in ET or NDWI is strongly affected by the vegetation status, i.e., vegetation coverage or vegetation index. For example, during the green-up period, both ET, GPP enhanced as a results of vegetation greening. However, lower ET or NDWI values in the earlier period does not indicate that vegetation is more water limited. Normalization is necessary to use these variables to assess water stress.

We agree entirely with the reviewer: ET and NDWI are strongly affected by vegetation status, and the low values in Green-up do not have the same meaning as the same low values during senescence for instance. But it is precisely because of this that the downscaling model is calibrated at every time step separately and independently, based on locally adjusted constraints. As a result, the NDWI and ET are effectively normalized as suggested by the reviewer. We have stressed this explicitly in the discussion of the revised manuscript.

2. As a journal specifically targeted at publishing dataset, I would suggest the authors provide enough details in the method for generating this SIF dataset. For example, in the last paragraph of section 2.4, how does the eigen decomposition work is not clear. The spatial and temporal window sizes are also not informed. Although the original method is described in details in a previous publication, since this journal is a data journal,the readers should gain enough understanding of the dataset without referring to other papers. Otherwise, this paper is more like an addendum or update to the previous paper.

The eigen decomposition is actually not part of the original downscaling approach, but rather part of the use of the index of agreement. This is a very technical procedure that would considerably overload the text and that is not necessary for the actual downscaling that is reported in the present data descriptor. As this decomposition is fully explained in the supplementary material of the dedicated paper (Duveiller et al. (2016) Sci. Reports.), which is in full open access, we think it is best that we redirect readers specifically to that document (i.e. section 5 of the supplementary information of that paper) instead of repeating everything here.

Regarding the operations specifically related to the downscaling, we will revise the text to ensure all necessary information is there. Regarding the spatial window mentioned by the reviewer, we thought what is already specified on page 6 line 4 to be enough: "… using an adaptable spatial moving window containing the 40 nearest observations around the central pixel". But following the recommendation we added more detail, such as the fact that those 40 observations need to be within a larger box of 11 by 11 GOME2 pixels.

3. The author mentioned that the dataset has spatial and temporal gaps in some areas due to the missing values for the GOME-2 SIF or the predictor variables. Would there be a method to solve this issue? The author mentioned about the potential usage for this dataset, however, the gaps would limit these potential applications.

We already dedicate a paragraph on discussing how the gaps could be filled in the current version of the manuscript (see page 11, lines 5 to 11).

4. The JJ SIF dataset shows an abnormal decreasing trend due to the sensor degradation (Zhang et al., 2018), how about the PK dataset? Since the downscaling are based on these two datasets, this needs to be further discussed. How does the algorithm deal with this issue, if the problem still exists, this needs to be informed and the users should be cautious for trend analysis using this dataset.

The trend mentioned by Zhang et al. (2018) should affect both JJ and PK datasets in the same way. We will mention this in the revised manuscript. Regarding how our algorithm deals with it, basically, the way our downscaling is parametrised (i.e. individually at every separate time step), the trends in the input SIF data should be reflected in the downscaled SIF data. Therefore, this is a problem of the GOME2 data in general, not specifically of our downscaled SIF product. A warning about this has been added in the revised manuscript.

Below are some detailed comments:

P3 L2 "land science"-> "Earth science"?

Changed

P6 L2, it would be good to explain the meaning of these b parameters a little bit, it will better help readers understand the ranges used in Table 1.

We have added a dedicated paragraph describing the meaning of these b parameters rigth after their introduction in equation (4).

P8 L4, do you have any references to support this? The two instruments should be exactly the same.

The instruments are the same but they are on different platforms (Terra vs Aqua) that are on different orbits (descending vs ascending) that have been in space for different amounts of time (since 2000 and 2002 respectively) and thus differently exposed to sensor degradation. All these differences can be reflected in the data. Regarding the specific point of degradation we will add the following reference:

Sayer, A. M., et al. "Effect of MODIS Terra radiometric calibration improvements on Collection 6 Deep Blue aerosol products: Validation and Terra/Aqua consistency." Journal of Geophysical Research: Atmospheres 120.23 (2015): 12-157.

P8 L5, why only on PKdata? What about JJ data?

These tests are relatively expensive from the computational side, as the entire dataset needs to be downscaled for three years everytime that a new combination if parameters is tested. We decided that, given the likelihood that the choice of the LST may be so relevant to improve the downscaling procedure, and that the TERRA observations may be of lower quality, we would look at this issue only for one of the two datasets.

We have reiterated this in the text, mentioning it for instance in the end of the sub-section 2.3 and later on in the results        .

P10 L30: I don't think so, this is just a high-resolution SIFdataset, it cannot be compared directly with TROPOMI SIF, for example, you cannot use downscaled SIF for year 2017 and compared with TROPOMI SIF for 2018 to detect changes.

If the actual change on the ground has an noticeable effect of the downscaling variables used (NIRv, NDWI or LST), we would expect to be able to see some change. A strong land cover change would probably be reflected for example. However, we agree that the downscaled SIF cannot fully replace a TROPOMI SIF retrieval. We have added a phrase to warn users about this point and further declared that the degree to which downscaled SIF can serve as a surrogate for TROPOMI will have to be further investigated once more data is available.

[revised manuscript text omitted]

---

## Author Response (AR2)

Dear Editor,

Please find below the responses to the minor revisions requested by the reviewers. To facilitate the reading you will find the reviewers comments in blue italic font, followed by our response in normal font. You will then find the revised version of the manuscript with annotated changed.

**REVIEWER 1**

*The manuscript is much improved compare to the previous version. I do have an additional minor comment. Currently the bias correction is not clearly described, is it a simple regression between the downscaled SIF and TROPOMI SIF in the time domain for each pixel? Some explanations in the method section would be helpful.*

The bias correction is done based on the systematic agreement quantification described in section 2.4. As mentioned there, the method is fully described in detail in section 5 of the supplementary information of the paper Duveiller et al. (2016). We have added a phrase to explicitely say that this was used to do the bias-correction to make sure it is clear.

**REVIEWER 2**

*High spatial resolution SIF is uniquely important for studying global photosynthesis, thus global carbon cycle and Earth system science as emergent new remote sensing technology. I am happy to see the publication of this dataset which is built on the top of an earlier published method, and with several follow-on work. The manuscript is well written and it well fit the scope of ESSD.*

*Two minor comments for the authors' consideration:*

*First, the authors used OCO-2 SIF as a benchmark to evaluate the downscaled dataset and Eq 2 was employed to make OCO-2 and GOME-2 SIF directly comparable. The scaling coefficients in Eq. 2 was determined by leaf-level measurements – do they change when upscaling SIF to the canopy scale?*

This is a good point. There might indeed be changes. However, we wouldn't be able to define a scaling approach that could account for all the canopy-scale variability that one can find globally. Furthermore, we think that the effect of canopy structure would be more important below 740 nm. Above that, in the NIR region (in which the OCO-2 bands are measuring), there is no pigment absorption and all leaves become very similar to each other. We therefore decided to use this leaf-level scaling that are based on very reliable measurements. We added a couple of lines to justify this in the text.

*I also suspect that there are BRDF effects on these coefficients due to the two sensors have different sun-sensor geometries.*

Since we use spatio-temporal averages (and not single soundings), the BRDF effects should be mitigated, as we are averaging multiple observations with different viewing-illumination geometries.

*Second, maybe I missed something, but I don't find empty triangles (replacing MYD LST with MOD product) in Figure 1 for JJ data? Please clarify.*

The empty triangles are not missing, but instead do not exist. As we mentioned in the text, for computational reasons we did not perform all the different variable permutations and gave priority to those that we expected would be more meaningful for our study. We added a line in the caption to clearly state that these triangles are not missing.

*Overall, I think this is a good article and should be published with a little more clarification. Congratulations.*

Thank you.

[revised manuscript text omitted]